# WACSAW: An adaptive, statistical method to classify movement into sleep and wakefulness states

Austin Vandegriffe[1]*, V.A. Samaranayake[1], Matthew S. Thimgan[2]*

**1** Department of Mathematics and Statistics, Missouri University of Science and Technology, Rolla, Missouri United States of America, **2** Department of Biological Sciences, Missouri University of Science and Technology, Rolla, Missouri United States of America

* austin.g.vandegriffe@gmail.com (AV); thimgan@mst.edu (MST)

## Abstract

Wearable actimeters can improve our understanding of sleep in the natural environments. Current algorithms may produce inaccuracies in specific individuals and circumstances, such as quiet wakefulness. New hardware allows data collection at higher frequencies enabling sophisticated analytical methods. We have developed a novel statistical algorithm, the Wasserstein Algorithm for Classifying Sleep and Wakefulness (WACSAW), to identify behavioral states from recordings of everyday movement. WACSAW employs optimal transport techniques to identify segments with differing activity variability. Functions characterizing the segments' movement distributions were clustered into two groups using a k-nearest neighbors and labeled as sleep or wake based on their proximity to an idealized sleep distribution. It returned >95% overall accuracy validated against participant logs in the test data and performed ~10% better than a clinically validated actimetry system. We present the methodology describing how WACSAW results in a novel, individually-tuned, statistical approach to actimetry that improves sleep/wakefulness classification and provides auxiliary information as part of the calculations that can be further related to sleep-relevant outcomes.

## Introduction

Sleep deprivation continues to be a pervasive problem with sleep time decreasing for individuals choosing other activities over sleep [1,2]. Inadequate sleep is increasingly linked to disease, including 7 of the 15 leading causes of death in the U.S. [3]. These conditions and harmful states include diabetes [4,5], cardiovascular disease [6–8], inflammation and immune dysfunction [9–12], and among other possible health complications, including earlier death [7,13–17]. It is also linked to cognitive decrements [18,19] that impact everyday task performance. Many of these studies are based on self-report and subjective sleep reporting which can be prone to errors

**Data availability statement:** The data and code are housed at the Missouri S&T Scholars' Mine, a permanent database https://doi.org/10.71674/mq6j-z250.

**Funding:** This study was funded by the Ozark Biomedical Initiative (#60544 to MST & VAS), the College of Arts, Science, and Education Best-In-Class funding from Missouri S&T (to MST), and the Leonard Wood Institute/Department of Defense (#W911NF2220200 to MST and VAS). The last sponsor supported the publication of this manuscript and played no role in any part of this study.

**Competing interests:** The authors have declared that no competing interests exist.

[20,21]. Moreover, estimates from laboratory studies may have the confounding factor that people sleep differently in real world environments than in a lab setting [22]. In addition, laboratory subject populations are often smaller and may not generalize to the broader population. Thus, objective recording in real world environments would provide practical insight into how natural sleep varies across individuals, manifests in clinical populations, and relates to increased likelihood of health disorders.

Wearables, such as wrist actimeters, have been employed by the sleep field for the last 40 years to attempt to gather objective sleep data in the natural environment. They have emerged as a practical method for determining sleep parameters, especially in long-term or field experiments where polysomnography (PSG), the gold standard for sleep detection, is impractical. In 2007, the American Academy of Sleep Medicine determined that actigraphy was accurate enough to determine total sleep time for both healthy individuals and individuals with sleeping disorders [23]. Thus, there is a general interest in having an automated system that can be deployed widely across the population and does not require a large expense of human effort.

Despite the successes of current actimetry, there remain acknowledged weaknesses in current data collection and analysis that limit the utility of these algorithms. Two of the most commonly employed approaches, the Sadeh [24] and Cole-Kripke [25] algorithms, were developed using regression methods. While these approaches have provided valuable insights into sleep biology, regression methodologies optimize results for population statistics that allows some individuals to deviate substantially from the overall average. It is unclear which individuals may deviate and requires human input to validate records against a reliable source. Additionally, these regression-based algorithms have difficulty identifying quiet wakefulness [26], possibly due to sampling rate, computational or statistical power. These algorithms have provided useful data for understanding variations in naturalistic sleep and continue to provide useful data. Yet, technological innovations in sensor, computer, and statistical technology allow for improvements in sleep and wakefulness categorization from real-life movement data.

More recently developed algorithms have applied machine learning techniques [27–37]. These approaches were not substantially more accurate than the regression-based algorithm results. For example, Van de Water and colleagues [26] applied an AI-based algorithm that yielded behavioral categorization with 86–99% accuracy for sleep sensitivity (i.e., the ability to detect sleep) but only a 27–68% accuracy for sleep specificity (i.e., ability to detect wake), likely due to misclassification of quiet wakefulness activities, which includes desk work, watching television, and reading [38]. When the performances of numerous popular algorithms were directly compared, the Sadeh algorithm was best for healthy individuals, even compared with AI-based methods [39]. The machine learning algorithms may suffer from a high sensitivity to the training set that does not generalize to variability of characteristics within a novel set of individual data [40–42]. Additionally, the GGIR algorithm has integrated new data collection techniques to estimate sleep from movement from higher frequency data collection [43–45] but leaves room for improvement and does not have any metrics that can be used for validation. Thus, there remains a need for

an improved actimetry algorithm that will be more automated, facilitate 24-hr recording and begin to adapt to individual activity profiles.

Here we present a novel algorithm, the Wasserstein Algorithm for Classifying Sleep and Wakefulness (WACSAW), that employs a statistical and data-driven approach to classify movement segments into sleep and wakefulness states based on raw activity data. WACSAW applies optimal transport and clustering methods to first segment movement profiles and then to assign sleep or wakefulness. It analyzes high frequency movement data collected using MEMS actimetry devices. When full day data were compared to a detailed activity log, WACSAW correctly categorized >95% of the states. We also compared WACSAW's performance with the commercial Philips Respironics' Actiware 6.0.9 algorithm on Actiwatch Spectrum Plus data [46], as well as a second comparison to GGIR. WACSAW performed better during quiescent activities and generates personalized results and minimizes both intra- and inter-individual variability of the results. With a median accuracy, sensitivity, and specificity >95% subject to detailed activity logs, WACSAW demonstrates the feasibility of a statistical approach to actigraphy. Thus, WACSAW is a novel algorithm that takes in higher frequency data and returns a more accurate categorization of sleep and wakefulness. Additionally, it is adaptive to individuals, contains explainable parameters, has interim metrics that can be used to potentially determine reliability of performance on individuals and has the potential to help in identifying other conditions within sleep based on movement recordings.

## Results

### Development and details of WACSAW

To develop WACSAW, we started with the movement data from 6 individuals ("Development Cohort", Table 1). This cohort spanned a large age range from 18–72 y.o. (mean of $42 \pm 19.1$ y.o.; 83% male; 66% white). Because of the limited sample size of participants, we included participants that detailed situations in their detailed logs that suggested situations of sleep and wakefulness patterns that posed known hardships to past analyses. For example, records show that participant data contained low activity wakefulness, such as reading in bed or watching TV, at least one instance of split sleep at night, and significant nighttime movement and potential sleep interruption. From the detailed participant logs, we chose to develop the algorithm on participants that reported (1) split nighttime sleep; (2) very low overall movement throughout the day and during waking activities; (3) older individual with more movement during sleep; (4) young with sporadic sleep patterns; (5) female with nighttime awakenings; and (6) significant nighttime movement that may be similar to numerous sleep disorders. We reasoned that if the algorithm could correctly classify these events, along with the more normal sleep patterns, then it would likely better generalize across a general population. These participants were used for all development sections and the algorithm was validated on a larger, independent cohort after development.

**Step 1: Data preparation.** The GENEActiv device uses MEMS to record component of gravitational force changes in perpendicular $x$, $y$, and $z$ directions where $x$ and $y$ are in the plane of the face of the watch and $z$ is perpendicular to that plane. Data were recorded at 10 Hz frequency resulting in a representative movement time series (Fig 1A). It has been determined that movement away from the direction of gravity sufficiently captures meaningful movement in all 3 directions and can be captured with the tilt angle time series [38]. The tilt angle reduces the dimension of data stream needed for developing the algorithm. Thus, data from the 3 directions were combined to determine the tilt angle ($\theta_t$) using equation 1:

$$\theta_t = \arccos\left(\frac{z_t}{\sqrt{x_t^2 + y_t^2 + z_t^2}}\right),$$

(1)

where $x_t$, $y_t$, and $z_t$ are the force changes in the $x$, $y$, and $z$ directions (axes) at time ($t$). The tilt angle preserves periods of high and low volatility (Fig 1A). In addition, there are periods that are in between high and low volatility (green label, Fig 1) that represents quiet wakefulness according to the activity logs.

 

**Table 1. Participant demographic data.**

| Participant # | Age | Sex | Ethnicity | Cohort in which data were used | | | | |
| | | | | Development | Validation | Independent | Actiwatch | GGIR |
|---|---|---|---|---|---|---|---|---|
| 1 | 24 | F | White | 1 | 2 | | 1 | |
| 2 | 47 | M | White | 1 | 2 | | 1 | 1 |
| 3 | 23 | M | White | 1 | 2 | | 1 | |
| 4 | 72 | M | South Asian | 1 | 2 | | 1 | |
| 5 | 26 | M | South Asian | 1 | 2 | | 1 | 1 |
| 6 | 60 | M | White | 1 | | | | |
| 7 | 47 | F | White | | | 1 | 1 (~58') | |
| 8 | 21 | M | White | | | 1 | | |
| 9 | 25 | M | White | | | 1 | 1 (~777') | |
| 10 | 30 | M | East Asian | | | 1 | 1 | |
| 11 | 40 | F | White | | | 1 | | |
| 12 | 25 | F | East Asian | | | 1 | | |
| 13 | 21 | F | White | | | 1 | | |
| 14 | 49 | M | Hispanic | | | 1 | | |
| 15 | 23 | M | White | | | 1 | | |
| 16 | 27 | F | White | | | 1 | 1 | 1 |
| 17 | 26 | M | White | | | 1 | 2 | |
| 18 | 50 | F | White | | | 1 | 1 (~230') | |
| 19 | 18 | M | White | | | 1 | 1 (~316') | |
| 20 | 18 | M | White | | | 1 | 1 (~144') | |
| 21 | 45 | F | White | | | 1 | 1 (~173') | |
| 22 | 47 | M | White | | | 1 | 1 (~58') | |

[1]Indicates where that one 24-hr dataset was analyzed for that participant

[2]Indicates when a second independent 2-day period of analysis was used

Parentheses indicate the minutes (') shift to align GENEActiv with Actiwatch start point.

**Step 2: Segmentation.** Higher frequency data captured by the MEMS sensors provide more opportunity to detect rapid changes in the tilt angle (Fig 1A). To isolate changes in the tilt angle irrespective of the absolute magnitude of force, we took the first difference of the tilt angle ($\Delta\theta_t$; Eq 2), which is the difference between adjacent observed angles $\theta_t$ and $\theta_{t-1}$ (Fig 1B):

$$\Delta\theta_t = \theta_t - \theta_{t-1} \tag{2}$$

Periods of high activity (and thus high volatility) will produce distributions of first differences with larger magnitude compared to those periods of lower activity. Notably distributions of the first difference with varying characteristics can be observed. For instance, participant logs confirm different activity periods, such as high (walking, Fig 1C), intermediate (reading in bed, Fig 1D), and low (sleeping, Fig 1E) in which the high active periods have distributions with a higher standard deviation compared to the quiescent and sleep period distributions. However, the difference in reading in bed (Fig 1D) and sleeping (Fig 1E) is subtle, with quiescent wakefulness exhibiting a noticeably higher variability relative to a validated sleep segment. To capture and emphasize such differences and to distinguish between states, we need a sensitive, yet robust method for comparing distributions. Going forward, we will refer to the first difference of the tilt angle (Fig 1B) as "movement" and the distribution of the first difference of the tilt angle (Fig 1C-E) as "movement distribution."

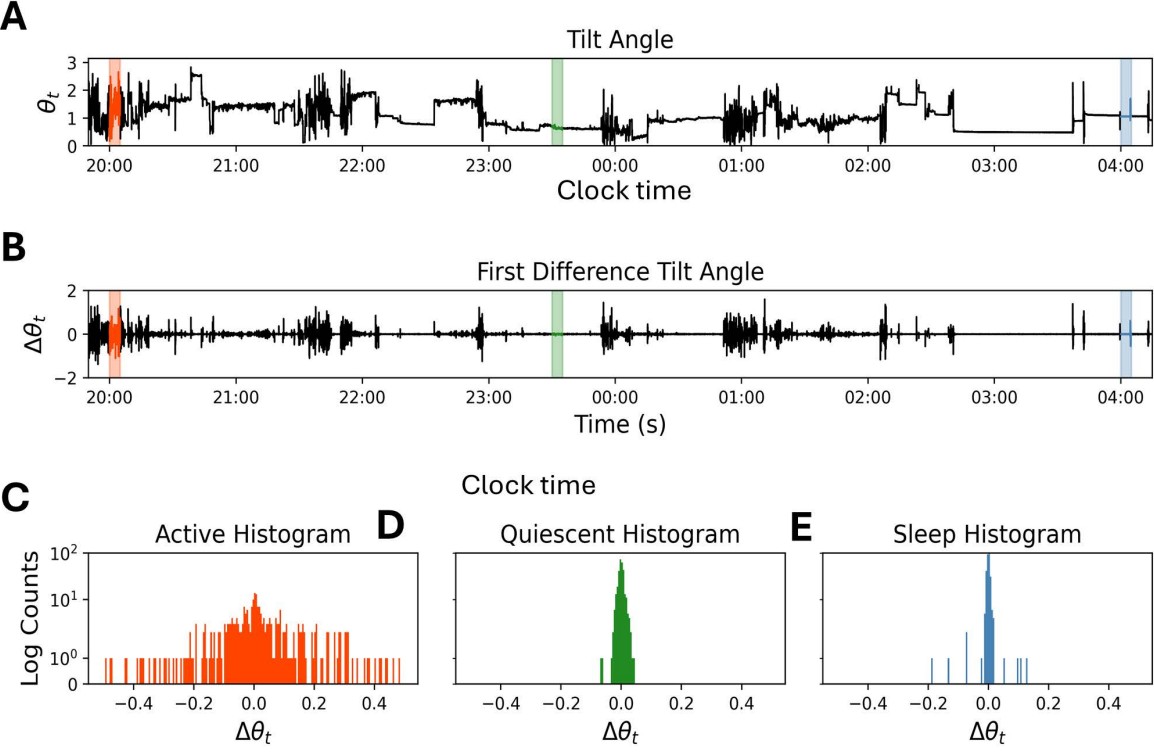

**Fig 1. Distribution of movement variability during different activity intensities.** (A) The tilt angle time series from 8 hours of activity recordings. These recordings have both a variation and intensity component. (B) First difference (FD) conversion of adjacent points of the raw tilt angle series from A ($\Delta\Theta_t$) isolates the movement variability from the intensity level. The orange background indicates an intense period of movement, the green indicates more quiescent wakefulness, and blue indicates sleep according to activity logs. (C-E) The first differences within these 5 min periods were plotted as a histogram in the corresponding colors to visualize the differences in movement. Note the similar nature of quiescent wakefulness (D) and sleep (E).

## The wasserstein time series and segmentation

The use of the Wasserstein metric, a scalar measure based on optimal transport theory, is well suited to distinguishing subtle distributional changes. The greater the difference between two distributions, the higher the Wasserstein distance between them. Mathematically, the Wasserstein distance is given by (Eq 3)

$$W^{(p)}(x, z) = \left( \frac{1}{n} \sum_{k=1}^{n} |x_k - z_k|^p \right)^{\frac{1}{p}}$$

(3)

where $x$ and $z$ are two time periods of the same length $n$, with the first differences of the tilt angle for the time period sorted in ascending order. We use the notation $x_k$ and $z_k$ to denote the $k$th element of the sorted periods ($x_k \leq x_{k+1}, z_k \leq z_{k+1}, k = 1, 2, ..., n-1$) and $p$ to denote the selected order of the Wasserstein distance. Note that in the current context, the elements $x_k$ and $z_k$ denote the ordered $k$th first difference tilt angle ($\Delta\theta$) corresponding to the two time periods that are compared. The value of $p$ was selected to be 1 and $n$ was taken to 5 minutes. Details about the order $p$ and $n$ can be found in the supplemental materials. A detailed treatment of optimal transport (OT) is presented in [47,48] and methods of computational OT in [49].

The Wasserstein distance, in our use case, converts differences between two movement distributions into a single score enabling us to detect time points at which movement distributions (and potentially the type of activity) changes.

Fig 2 illustrates how this is achieved. Given a 1 second point of interest (arrow in Fig 2A), WACSAW compares the movement distribution of the 5-minute interval immediately prior to the reference point, to that of the interval five minutes after it. Fig 2A present a 20-minute example of movement with an abrupt transition from wakefulness to sleep at the reference point. The Wasserstein distance between the movement distributions of the two intervals is computed and is referred to as the Wasserstein score for that time point. Wasserstein scores are computed for each second of the movement time series. This resulting series is called the Wasserstein Time Series (WTS).

Fig 2B illustrates a more complex example of the construction of the WTS where the movement transitions from a sleep period to wake and then back to sleep. The data in Fig 2B is truncated, and values before and after the presented time interval were used in calculating the Wasserstein distance. As the algorithm approaches the transition from low variability to high variability, the Wasserstein score (purple curve) increases as the differences in distributions increases. As the time series moves into the middle of the more volatile period, the Wasserstein score decreases as the forward- and backward-looking distributions appear more similar. As the volatility decreases in the forward-looking period and the backward-looking period remaining volatile, the Wasserstein score increases again. Lastly the Wasserstein score settles near zero as the volatility becomes very low in both the forward- and backward-looking series when the point of interest moves into the period of low volatility.

We next identify time points where possible activity changes occurred. These change points are referred to as segment boundaries. To define a segment boundary, we determined the peak of the Wasserstein series as the cut point (vertical red lines in Fig 2B). Yet, all peaks in the WTS do not denote activity changes and would result in too many segments as well as segments of very small duration. Thus, we employed a detection algorithm that only identified peaks above a threshold which we termed the Change Point Threshold (CPT; horizontal, dashed red line in Fig 2B). The WACSAW algorithm is applied to an individual's tilt angle time series with the CPT computed over disjoint 2-day segments (see hyperparameter optimization in the SA Appendix for a detailed explanation and S1 Table). Not only is the CPT unique for that individual but also allows it to vary over longer periods for that individual because of the re-computation of the CPT after two days. We designate the region of contiguous time points above the CPT to be the change point region, the peak detection algorithm was applied within each change point region (gray regions in Fig 2B). Thus, within each such region, WACSAW determines and labels the highest Wasserstein score within that region as the change point (vertical red lines).

How sensitive the Wasserstein distance is to differences between distributions can be controlled by the distance order $p$, lower values magnify differences between distributions. Thus, an appropriately chosen value of $p$ will enable us to distinguish differences between distributions associated with quiescent wakefulness and sleep. In addition, the use of the individual-specific CPT combined with the chosen sensitivity of the $p$ value in the Wasserstein equation can be used to tailor WACSAW to individual data, behavioral patterns, and research questions. In contrast, use of a population-based threshold that signifies sleep periods below it and wakefulness above it would not be adaptive to individual differences.

With this change point detection procedure, we identified regions within which relatively homogeneous volatility is detected, especially in the waking period. However, the segmentation protocol produced too many small regions, with too little data to analyze for classification because of the highly sensitive nature of the Wasserstein algorithm we employed. For segments less than 3 secs, the segment was automatically concatenated to the previous segment. For longer segments, we sought to merge adjacent regions with relatively similar volatility. We used the nonparametric Levene Test for Equal Variance [50] to compare the distributions of first differences, independent of length, between adjacent segments. For segments longer than 30 seconds, where there may be increased occurrence of outliers, the following strategy was employed to eliminate the influence of outliers on the Levene Test. We took the absolute value of the first differences of the tilt angle and eliminated the extreme 25%. From the remaining first difference values, we calculated the mean and standard deviation and further eliminated data points outside 1 standard deviation from the mean. This data filtration is not practical for segments between 3 and 30 seconds due to limited availability of data within the segment, and thus, the Levene Test was applied directly. If the current segment was not significantly different from the previous segment, at $a = 1e^{-25}$,

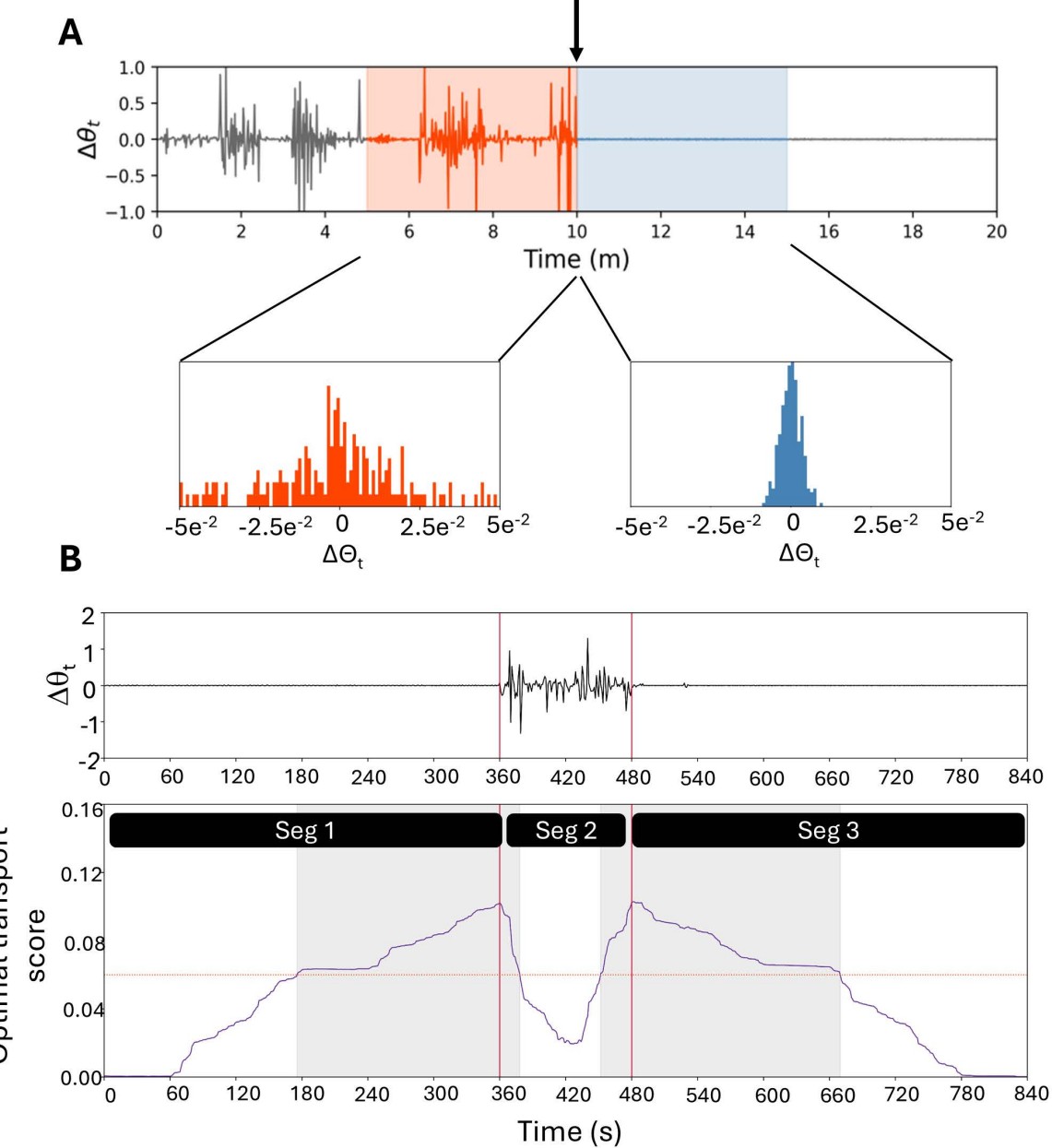

**Fig 2. Example of the strategy for obtaining segments within an activity time series.** (A) To determine the optimal transport score for a given point (vertical arrow), we compared the distribution of the FD of the tilt angle for each second from the 5 minutes prior (orange shaded area) to the distribution of FD in the 5 minutes after the point of interest (blue shaded region). The distributions for this example are shown in the bottom half of the panel. (B) In a magnified example of where the segmentation points are selected, there is a quiet period (Seg 1) next to a short active period (Seg 2), and then the participant returns to a quiet period (Seg 3). Because the quiet period of Seg 1 is longer than 5 min, there is a very small score at the beginning of the Wasserstein scores. As the WACSAW calculation moves into a region of larger difference, the Wasserstein score increases and crosses a threshold (horizontal orange stippled line; see text for details on how it is calculated). The whole area above the threshold is designated with a gray background. From this region, the peak of the Wasserstein score is identified and marked as the changepoint. As WACSAW moves through the area in Seg 2, the distributions become more similar even as they are more volatile, and the score decreases. As the calculation moves into the second quiet period, the Wasserstein score again increases as the distributions become more different again. The score crosses the threshold and the peak is again determined. Thus, from this small time series, WACSAW identified 3 segments corresponding to the activity plot in **(B)**.

the two segments were merged. To achieve proper computation of such small values, we used arbitrary precision floating point arithmetic. Though the significance level employed for joining the segments seems extraordinarily strict, it provided the optimal discrimination between segments (see hyperparameter section for a discussion). If the segment of interest was not different from the previous segment, the previous and the current segments were merged. If the current segment was significantly different from the previous segment, then we compared it with the subsequent segment. If the Levene's Test did not reach the difference threshold, they were merged and if they were different, they maintained designation as independent segments. This process reduced the number of segments to be classified. For example, this process resulted in a reduction of segments from 716 to 118 in a 2-day recording, which was typical of the results we analyzed.

We use Fig 3 to illustrate the consequences of the Levene Test. In Fig 3A, each vertical line represents a change point defined by the initial WACSAW algorithm. The vertical dotted red lines in Fig 3B represent those change points with insignificant difference in variation on either side as determined by the Levene test. Thus, WACSAW removes the segmentation boundary and the distributions of the segments on either side of the boundaries are merged. In Fig 3C, the four initial

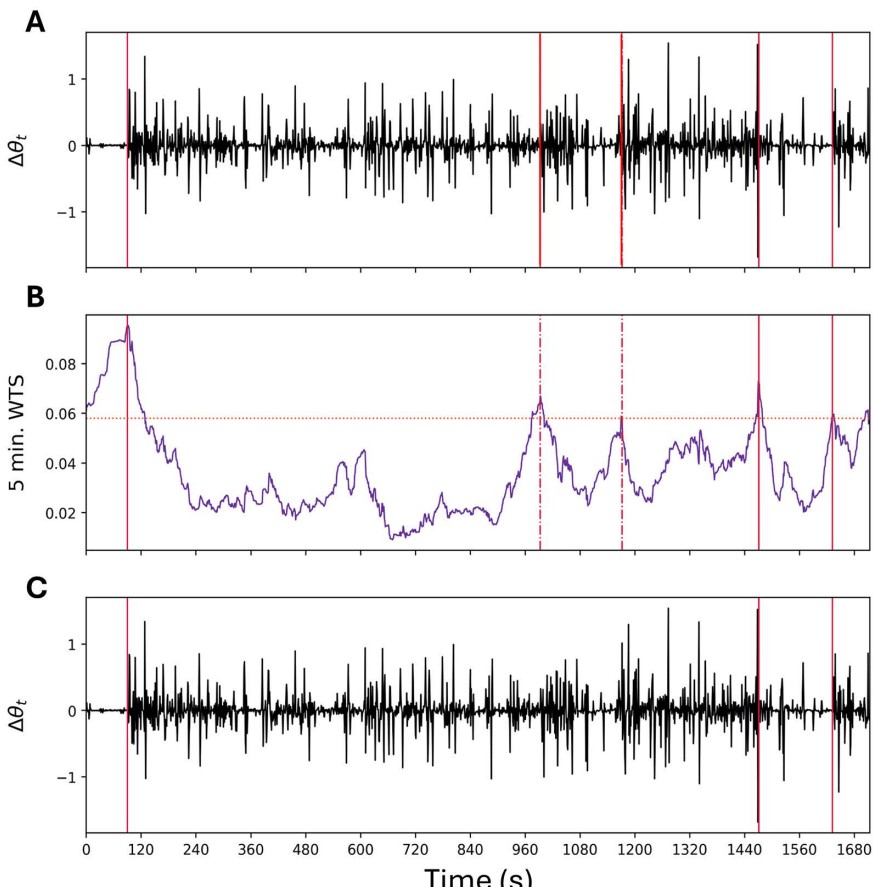

**Fig 3. Example of the implementation of the Levene Test combination of segments.** (A) To determine whether two segments should be merged, WACSAW starts with the first differences plot for a given segment. The vertical red lines are a reference for where WACSAW assigns segmentation divisions from data in (B). (B) WACSAW converts the first differences data into an optimal transport score time series (WTS = Wasserstein Time Series). Each segment is then compared to the previous and the subsequent segment using the Levene test to determine if segments should be merged. In this example, the horizontal dashed orange line represents the change point threshold. After the Levene test is applied, two of the change points between segments were merged (dashed vertical red lines). (C) The result of the Levene Test in which the final segmentation points identified by WACSAW are illustrated by the solid, vertical red lines.

segments were merged to form two segments as the segment boundaries defined by dashed lines were eliminated. This approach will ignore brief changes that may disrupt a long period of consistent behavior, such as a rollover event. It is typically robust enough to segment out brief awakenings with goal-directed behaviors, such as a short visit to the bathroom in the middle of the night. These segments will be subject to sleep-wake classification.

## Segmentation performance

We evaluated the results of the segmentation portion of WACSAW across the six subjects on whom we trained the algorithm. These 6 participants had features that posed challenges to other algorithms. There is no absolute comparator that quantifies the quality of segmentation, though we will show the results of the algorithm's performance later in the manuscript, which depend on proper segmentation. As expected, there are more frequent cuts during daytime wakefulness compared with sleep periods because waking activities vary throughout the day. Segments are of varying lengths which also would be a hypothesized property of daily behaviors. One overnight example demonstrates the general performance of the segmentation portion of WACSAW (Fig 4). WACSAW identifies segmentation points at what looks like extended periods of movement volatility, including what appears to be different levels of waking activities from 8–9 AM (Fig 4A) and a short waking period between 3 and 4 AM. On the other hand, WACSAW does not identify infrequent and brief movements throughout the night as independent change points. These brief movements were not reported in the logs and may reflect quick postural shifts, such as roll over events. The quick shifts did not elevate the Wasserstein score above the change point threshold, and thus these movements are not designated as separate segments according to WACSAW.

**Step 3: Classification.** The last step is to classify the segments into sleep and wakeful states. The classification of segments is done in two steps: (A) initial separation of segments into potential sleep and wakefulness states, and then (B) a final classification utilizing the initial groups derived in step A. WACSAW uses that individual's activity data to define sleep and wakefulness states for classification of that individual's recording, which contributes to the personalization of WACSAW's analysis.

In Step A, we compared the movement distribution of each segment with an idealized sleep segment in which the movement values are all zero. To determine the values for an independent segment, one thousand subintervals of random lengths and from random starting locations within the segment were generated. We filtered the data by eliminating the extreme 5% of first differences to avoid misclassification of short sleep segments due to the presence of a few large movements. Fig 4B illustrates this filtration with the eliminated first differences shown in red. The Wasserstein distance between the distribution of the middle 95% of the first differences in each subsegment distribution and the idealized sleep distribution (baseline distribution degenerate at zero) was then computed (Fig 4B). This procedure generates one thousand Wasserstein scores for each segment, and these scores form a distribution reflecting the difference of a segment from that of idealized sleep (Fig 4C). For each segment, there are scattered values to the left and right of the primary peak, indicating subsegments with more or less variability outside the prominent peak. Fig 4C is an example of the distribution of the 1000 Wasserstein scores calculated from the subsegment scores with the most frequent value around 0.0075. This distribution is typical of one in which there are short bursts of activity within a larger low-variability segment. Mathematically, the Wasserstein distance used in this step of classification reduces to

$$D_i^{(l)} = \left( \frac{1}{n_i^{(l)}} \sum_{t \in S_i^{(l)}} \left| \Delta\theta_t^* \right|^k \right)^{\frac{1}{k}},$$

(4)

where $k$ is the order of the distance, $\ell = 1,\ldots, L$ represents the subsegment of interest, $S_i^{(l)} \subset S_i$ represents the time points within a subsegment of segment $S_i$, $n_i^{(l)}$ represents the number of points in the subsegment $S_i^{(l)}$. Note that $\Delta\theta_t^*$ is the first difference at time $t$ if the first difference is withing the middle 95% of the data in the segment and zero otherwise.

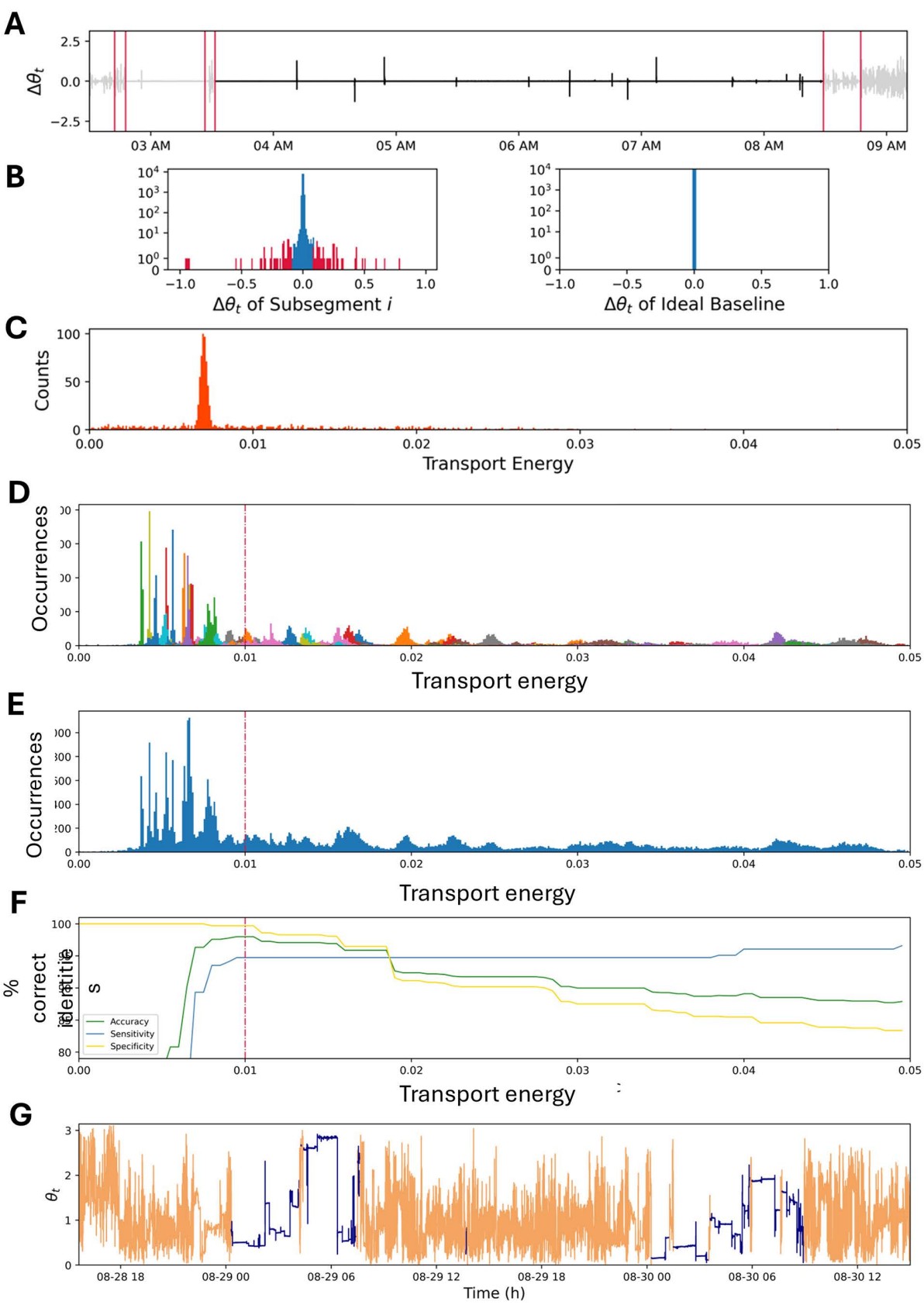

**Fig 4. Example of generating and defining the meaning of Wasserstein scores from first differences from movement variability.** (A) The first differences time series for a select segment with the resulting segmentation boundaries after the Levene test step designated with red lines. The segment being analyzed is in black, while grayed out segments will be treated similarly in an independent calculation. (B) Each of the randomly generated segments is individually compared to "idealized sleep" by optimal transport methods. The left histogram is the distribution of first difference values from the selected subsegment where we remove the extreme 5% of points (red bars are censored, blue bars are kept, note the logarithmic scale for the counts). That distribution is compared to the "idealized sleep" distribution in the right distribution to generate an optimal transport score. (C) The histogram of the optimal transport score for each of the 1000 iterations of the selected subsegment to characterize the variability of the entire segment, which appears to have a peak around 0.075. There are portions of the segment that have higher and lower variability with scores greater and less than 0.075, but they are a small minority of subsegments. The overall shape of the distribution characterizes the variability of the segment. (D) Histograms of the Wasserstein scores for segments derived from 2 days' worth of data from Participant 5. Different colors represent subsegments Wasserstein scores from different segments compared to "idealized sleep." Due to limited palette selection, some colors are infrequently duplicated. (E) The histograms for an individual participant for two days' worth of data without identifying individual segments. By eye, there appears to be a difference in the appearance of histograms below a 0.01 score and those above (represented by the vertical red line). (F) We iteratively tested the % of correct categorization using thresholds of Wasserstein scores from 0 - 0.05 compared against the activity logs. We evaluated overall accuracy (green line), sensitivity to correctly identifying sleep periods (yellow line), and selectivity to correctly identifying wakefulness periods (green line). In this example, the best balance of all three statistics was at a Wasserstein score of 0.01 (vertical red line). (G) Results of using a hard 0.01 threshold to classify sleep and wakefulness overlaid on the tilt angle data gathered for that time series. Orange indicates wakefulness calls and purple indicates sleep periods, as identified by WACSAW. The overall accuracy is > 90% under these conditions.

Borrowing from optimal transport phraseology, the value $D_i^{(\ell)}$ will be referred to as the transport energy associated with the $\ell^{\text{th}}$ resampled subsegment in segment $i$. The collection of $\left\{D_i^{(\ell)}\right\}_{\ell=1}^{L}$ generates an empirical distribution, denoted by $D_i$, for each segment. The value of $k$ sets the scale of Fig D and E and was chosen to be 20 based on visual inspection of development set, with selection based on the overlap of the transport distributions. Lower values of $k$ results in heavily overlapped distributions.

Over a full 2-day recording period, WACSAW generates a profile typical of the one presented in Fig 4D. Histograms from individual segments are differentiated by color. There is a cluster of tall, thin histograms to left side of the figure indicating less difference from idealized sleep. Moving to the right of the distribution profile, the peaks become flatter and broader, indicating more variability throughout the segment. This is usually an indicator of some wakeful activity, where subsegments within an activity segment can have greater variability than in a sleep segment. Note that there are values to the right of 0.05, but the graph has been truncated to highlight the distributions from 0–0.01 of the profile.

Based on observations from Fig 4E, which shows the cumulative Wasserstein score histogram that includes all occurrences at a given Wasserstein score from all subsegments, we hypothesized that sleep-like segments were to the left side of the graph while waking segments tended to reside on the right side of the graph. To test this hypothesis, we used a sliding cutoff point to determine if there was a threshold at which the accuracy of sleep and wakefulness calls were optimized. We tested this for each individual in the Development Cohort, and a representative example from one participant is shown in Fig 4F. The correct designations for each second of the recording were calculated as a percent of total recording time (accuracy), percent of correct sleep calls over total amount of sleep (specificity), and percent of correct wakefulness calls of the total amount of wakefulness (sensitivity). WACSAW calls were validated using the participant logs or adjusted log as the comparator group for what occurred at that time. As the Wasserstein score went from 0 up through 0.01, the accuracy of each of the measures improved, except for sleep sensitivity because when the program calls everything sleep it will always get the sleep segments correct. The optimal value appeared to be 0.01 for this individual, which resulted in the calls in Fig 4G. For this individual dataset, it resulted in a > 95% accuracy across the 2 days. Yet, this hard cutoff did not work for all individuals, though it was found to be between 0.01 and 0.02 for each individual in the development cohort. We also observed several shallow drops in accuracy between 0.01 and 0.02 (Fig 4F) indicating an overlap of sleep and wakeful segments because there was not a threshold that optimized the sleep and wakefulness calls for every individual. For the example participant, setting the threshold to a transport energy to 0.01 resulted in a > 97% accuracy (Fig 4F). But in other participants, a threshold of 0.01 reduced the accuracy, suggesting that an absolute population-based threshold was impractical. However, two cut-off values can be identified based on 2-day recoding periods for all individuals in the

development cohort, with one value (0.01) almost always containing sleep segments to the left of it, and the other value (0.02) almost always containing wake segments to the right of it. We took the average of these two cut-off values, 0.015, as a starting point in segment classification.

To enable the comparison of segments of unequal lengths and to capture all the information present in each of the movement distributions, we employed a process that transformed the movement distribution from each segment into a single curve, by obtaining the modulus of the characteristic function of that distribution. For a given segment $S_i$, the characteristic function $\varphi_i$ of $\Delta\theta$, the first difference of the tilt angle is computed using the formula

$$\varphi_i(\xi) = \frac{1}{n_i} \sum_{t \in S_i} e^{-j\xi(\Delta\theta_t)},$$

(5)

where $\underline{Si}$ is the $i^{th}$ segment, $n_i$ is the number of observations in $S_i$, $j$ is the imaginary unit $\sqrt{-1}$, and $\xi$ a real number, with $\Delta\theta_t$ defined as before. Since the domain of the characteristic function is the same for all segments, irrespective of segment length, the use of the characteristic functions facilitates comparison and classification across all segments regardless of size. For example, a 7 hr segment of one time series recording (Fig 5A), there are examples of highly variable segments (orange, Fig 5B), subtly variable segments (green, Fig 5C), and low variable segments (blue, Fig 5D) and activities were validated by participant log. The distribution data are transformed by Eq 5 that resulted in the modulus of characteristic functions, $|\varphi_i|$, and plotted (Fig 5E-G).

The modulus of the characteristic function (MCF) obtained from a given segment can be thought of as a curve that captures the movement patterns within that segment. Thus, the MCF curves from segments that are most likely sleep should cluster together and those from segments that are most likely from wakeful periods should cluster together as well. However, there are segments that will lie in between. To classify such segments, the following approach was taken. The modulus of the characteristic functions derived from segments with mean transport energy are averages less than 0.015 (likely sleep) are averaged to nucleate a cluster and modulus characteristic functions derived from segments with mean transport energy greater than 0.015 up to 0.1 (likely wakefulness; Fig 4F) are averaged to nucleate the second cluster. These two nucleated functions (or curves) are taken as clustering objects. Characteristic functions of segments with transport energy greater than 0.1 were not used because of the highly variable nature of these functions deteriorated classification accuracy. K-nearest neighbors clustering was then conducted on all the characteristics functions of an individual. Segments whose characteristic function clustered with the nucleated function associated with mostly sleep were classified as sleep. That is, segments were given the same label as the closest nucleated function (Fig 4F). Note that while the 0.015 threshold was determined universally, the nucleated clusters for sleep and wake are driven mostly by an individual's movement patterns. For example, using the 0.015 transportation energy threshold as the sole classification criteria did not produce same high accuracy that the characteristics function clustering approached yielded. Transport energy is a crude measure of how far or close a segment is to idealized sleep while the characteristics function captures complete information about a segment's movement distribution. In addition, the classification into sleep and wake is based on how characteristics functions associated with segments cluster specific to each individual. Thus, the clustering and resulting classification is to a great extent individually tailored.

## WACSAW performance across training and test cohorts

In the above sections, we have described how WACSAW was developed, the steps it uses to distinguish sleep and wakefulness from activity data, and provided some examples of how the process operates. Now, we present how WACSAW performs across multiple individuals from different experiments.

Within this Development Cohort, WACSAW performed admirably (Table 2). WACSAW classifies sleep or wakefulness on a second-by-second basis and produces results without human intervention. As covered in the Materials and Methods,

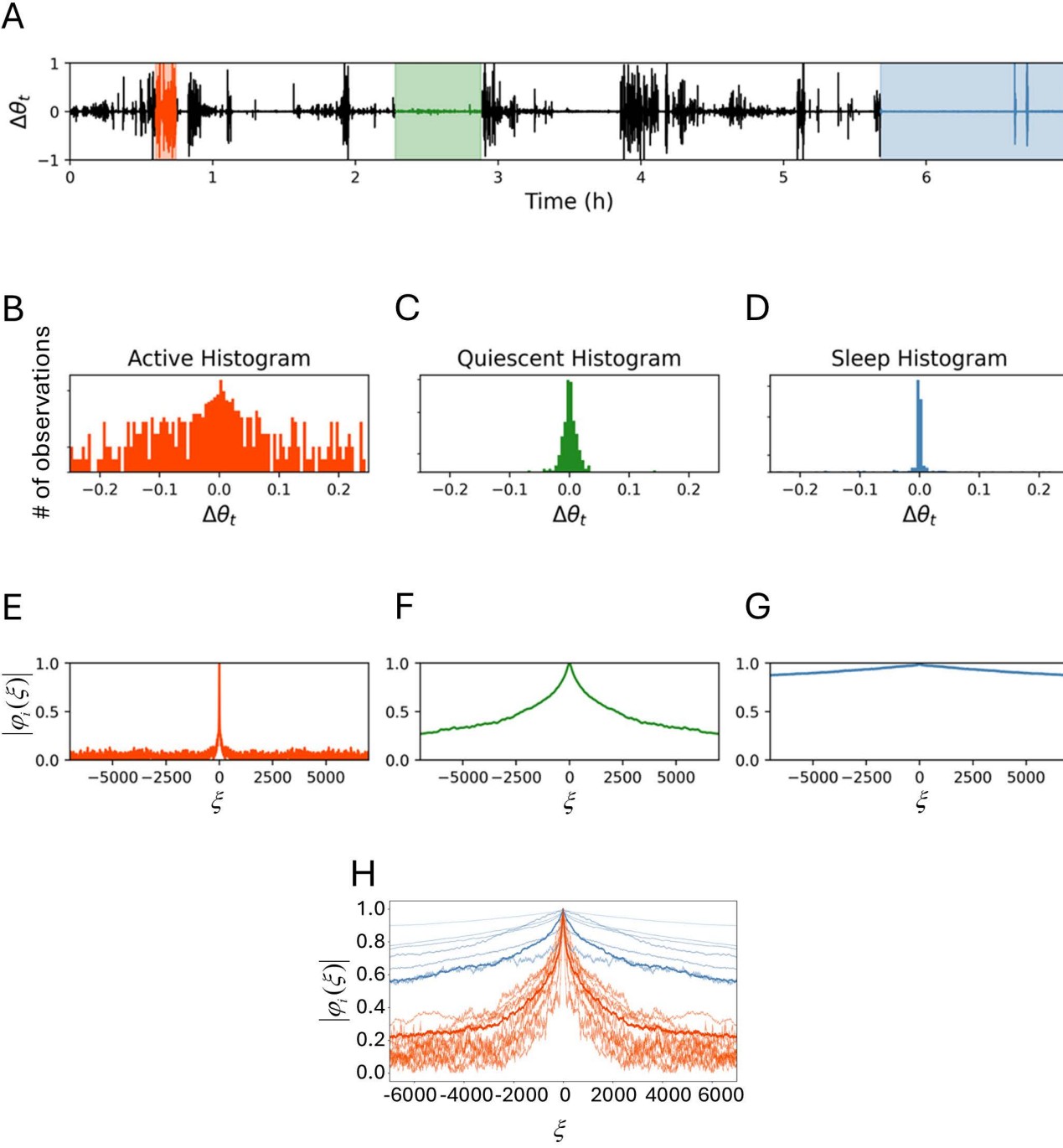

**Fig 5. Example of the conversion of first differences time series of the tilt angle into characteristic functions.** (A) A 7-hour segment of first differences data with examples of high variability, moderate variability, and low variability segments as noted in the participant's log. (B-D) Histograms of the distributions from the segments of the corresponding activity levels. The color of the histogram aligns with the background color behind the sub-segment of data graphed. (E-F) Each distribution is transformed into its characteristics function (CF) and its modulus (MCF) can be compared to one another in their shape and values at respective points along the graph. (H) is an example of the clustering of characteristic function moduli to classify sleep or wakefulness. Segments defined by WACSAW are classified as sleep or wakefulness based on the clustering of the MCF's. Each line represents a different segment. The bold blue and orange lines are the averages that nucleate the binary clusters. The average characteristic functions of segments with a Wasserstein score below 0.015 is calculated to nucleate the sleep cluster (thick blue line). The average characteristic function of segments with a Wasserstein score from 0.015-0.1 nucleates the wakefulness cluster (thick orange line). The characteristic function from each segment is then clustered using these averages as the anchor. Segments that cluster with the wakefulness characteristic function are classified as wake (lighter orange lines) and those that cluster with the sleep characteristic function are labeled sleep (lighter blue lines).

**Table 2. WACSAW results from the "Development cohort" in percent correct.**

| | Participant id | % correct Accuracy (raw) | % correct Sensitivity (raw) | % correct Specificity (raw) | % correct Accuracy (adj) | % correct Sensitivity (adj) | % correct Specificty (adj) |
|---|---|---|---|---|---|---|---|
| Development Group | 1 | 96.9 | 94.6 | 98.2 | 95.8 | 94.3 | 96.7 |
| | 2 | 96.7 | 95.9 | 97.0 | 98.8 | 98.7 | 98.9 |
| | 3 | 99.1 | 98.4 | 99.5 | 99.5 | 99.5 | 99.5 |
| | 4 | 96.0 | 93.3 | 97.5 | 97.6 | 96.8 | 98.0 |
| | 5 | 95.5 | 96.5 | 95.1 | 95.8 | 97.2 | 95.3 |
| | 6 | 96.5 | 99.0 | 95.5 | 98.4 | 99.0 | 98.1 |
| | Mean | 96.8 | 96.3 | 97.1 | 97.7 | 97.6 | 97.7 |
| | Median | 96.6 | 96.2 | 97.2 | 98.0 | 97.9 | 98.1 |
| | Std Dev | 1.1 | 2.0 | 1.5 | 1.5 | 1.9 | 1.5 |
| | MAD | 0.8 | 2.0 | 1.5 | 1.1 | 1.1 | 1.2 |
| | min | 95.8 | 93.3 | 95.3 | 95.8 | 96.8 | 95.3 |
| | max | 99.1 | 99.0 | 99.5 | 99.5 | 99.5 | 99.5 |
| | % > 95% | 100 | 66.7 | 100 | 100 | 83.3 | 100 |
| Development Validation | 1 | 92.4 | 94.7 | 91.3 | 94.4 | 97.3 | 93.0 |
| | 2 | 97.2 | 93.5 | 98.8 | 98.1 | 95.3 | 99.2 |
| | 3 | 96.6 | 99.4 | 95.3 | 96.4 | 99.6 | 95.0 |
| | 4 | 94.4 | 91.2 | 96.4 | 96.8 | 99.0 | 95.7 |
| | 5 | 95.1 | 93.7 | 95.9 | 96.7 | 97.7 | 96.0 |
| | Mean | 95.1 | 94.5 | 95.5 | 96.5 | 97.8 | 95.8 |
| | Median | 95.1 | 93.7 | 95.9 | 96.7 | 97.7 | 95.7 |
| | Std Dev | 1.9 | 3.0 | 2.7 | 1.3 | 1.7 | 2.2 |
| | MAD | 1.4 | 2.3 | 2.0 | 1.9 | 2.3 | 1.7 |
| | min | 92.4 | 91.2 | 91.3 | 94.4 | 95.3 | 93.0 |
| | max | 97.2 | 99.4 | 98.8 | 98.1 | 99.6 | 99.2 |
| | % > 95% | 60.0 | 20.0 | 80.0 | 80.0 | 100.0 | 80.0 |

we have compared the output to both the raw log entries from the participants (raw) as well as to an investigator adjusted log that was defined by a set of rules (adj). We present both sets of data for comparison and note that the adjusted data set results in only a marginal improvement, but they do not rely on the accuracy of than participant recollection. The overall accuracy of WACSAW is 96.8% correct classification with the raw data and 97.7% in the adjusted data. The correct classification of sleep episodes (specificity) is above 96% across this set and the correct classification of wakefulness (selectivity) is above 97%. The similarity of the median value reflects that there are unlikely to be single values that are responsible for pulling the mean in one direction or the other. In addition, the standard deviation (SD) and median absolute deviation (MAD), which is the average median of the absolute deviations from median across the group, suggest that the results are accurate consistently across individuals. Given the age and situational differences, it suggests that WACSAW can at least partially adapt to individual circumstances. The minimum and maximum accuracies for this cohort are all above or near 95%, which suggests consistency within the group that WACSAW was trained on. A high percentage of the values within the cohort are above 95%, suggesting very good classification. Given that the raw values and adjusted values are very similar and that log recordings are subject to errors because participants may not or are unable to document specific times for changes in activity, we will only discuss the adjusted values, but the raw data will be presented for the performance of WACSAW.

Five out of the 6 participants within the Development cohort had a second data collection period that was independent of the original data collection period on which WACSAW was developed. As an initial test of WACSAW's performance, we

applied WACSAW to this independent data collection period in the participants with the same properties as the Development cohort. Overall accuracy, specificity, and selectivity were at or above the 95% accuracy threshold for the entire group with only 1 participant records below this threshold for the 48-hour recording period (Table 2). The SD and MAD remain low across this small population. This test shows that the performance of WACSAW is not totally reliant upon the training data and can perform well on independent data, albeit obtained from the same subjects used in the development cohort.

To test whether WACSAW can perform equally well on independent data from individuals without the same underlying properties, we applied WACSAW to 48-hour recordings from 16 independent individuals (Table 3). Across the "Independent validation" data set, WACSAW maintained its average of greater than 95% accuracy across overall performance, sensitivity, and selectivity. The std dev and MAD remained low. There is one individual, participant 11, that had one day in which WACSAW demonstrated a significant underperformance. In a *post hoc* interview, it was determined that the person was sick over the data collection period. It highlights that WACSAW has room for improvement under specific circumstances. Under these circumstances, the interim metrics, like the cumulative histogram of transport energy discussed in Fig 4, may indicate that results are clear and reliable or that are not as delineated and may require further investigator validation. Despite the 24-hr underperformance, WACSAW performance in all individuals were above 86% with an overall average over 95%.

It is possible that we have been evaluating a set of data for which is it simple and easily definable to identify sleep and wakefulness. Therefore, we compared WACSAW's output to the sleep and wakefulness designations produced by the clinically validated Actiwatch that is produced and maintained by Respironics. The Actiwatch is a popular, validated

**Table 3. WACSAW performance in the "Independent cohort" in percent correct.**

| | Participant id | % correct Accuracy (raw) | % correct Sensitivity (raw) | % correct Specificity (raw) | % correct Accuracy (adj) | % correct Sensitivity (adj) | % correct Specificty (adj) |
|---|---|---|---|---|---|---|---|
| Independent | 7 | 97.7 | 94.2 | 99.4 | 99.4 | 99.2 | 99.5 |
| Validation | 8 | 96.6 | 94.4 | 97.9 | 97.7 | 97.3 | 98.0 |
| | 9 | 98.8 | 97.1 | 99.6 | 98.8 | 97.1 | 99.6 |
| | 10 | 97.3 | 94.8 | 98.2 | 97.2 | 94.7 | 98.1 |
| | 11 | 85.3 | 68.4 | 96.0 | 86.3 | 69.4 | 97.0 |
| | 12 | 98.9 | 98.3 | 99.2 | 98.9 | 98.4 | 99.1 |
| | 13 | 96.6 | 98.6 | 95.4 | 98.4 | 99.5 | 97.7 |
| | 14 | 93.2 | 99.6 | 91.2 | 94.4 | 99.9 | 92.6 |
| | 15 | 95.5 | 88.1 | 100.0 | 98.6 | 97.2 | 99.3 |
| | 16 | 86.4 | 98.3 | 82.4 | 98.8 | 97.2 | 99.7 |
| | 17 | 98.8 | 97.9 | 99.2 | 98.7 | 98.0 | 99.2 |
| | 18 | 97.6 | 98.8 | 96.9 | 98.8 | 99.3 | 98.5 |
| | 19 | 99.4 | 99.1 | 99.5 | 99.1 | 97.7 | 99.7 |
| | 20 | 98.9 | 96.8 | 100.0 | 98.9 | 97.0 | 100.0 |
| | 21 | 94.8 | 96.7 | 94.1 | 97.0 | 98.5 | 96.3 |
| | 22 | 86.4 | 91.8 | 84.0 | 89.9 | 95.5 | 87.3 |
| | Mean | 95.1 | 94.6 | 95.8 | 96.9 | 96.0 | 97.6 |
| | Median | 97.0 | 96.9 | 98.1 | 98.7 | 97.5 | 98.8 |
| | Std Dev | 4.8 | 7.6 | 5.5 | 3.7 | 7.2 | 3.3 |
| | MAD | 3.3 | 4.5 | 4.0 | 2.5 | 3.5 | 2.1 |
| | min | 85.3 | 68.4 | 82.4 | 86.3 | 69.4 | 87.3 |
| | max | 99.4 | 99.6 | 100.0 | 99.4 | 99.9 | 100.0 |
| | %>95% | 69 | 63 | 75 | 81 | 88 | 88 |

actimeter which uses piezoelectric accelerometers to obtain movement counts which are then used to determine sleep and wakefulness. For a detailed treatment of the Actiwatch algorithm, we refer to you to the Actiwatch documentation [46]. The data from these samples was analyzed in earlier charts but had not been benchmarked to an activity device that has been validated and approved for use in a clinical and research setting. For this period of data collection, participants were wearing both watches at the same time on the same wrist. The data from the Actiwatch was processed through the company's analysis software to obtain the classifications whereas WACSAW was applied to the GENEActiv MEMS data as described above.

We maintained the same divisions in the population because the first 5 participants were the majority of the data on which WACSAW was originally developed. WACSAW performs well on the data on which it was optimized as discussed above (Table 4), and it outperforms the Actigraph by roughly 10% on measures of accuracy, sensitivity and selectivity. Across the population, WACSAW performs better between individuals as the Std dev and MAD are much lower with WACSAW designations than with Actiwatch classifications. Nearly all the values for the WACSAW designations are greater than 95% accuracy in each of the criteria, whereas a minority, if any, of the Actiwatch cross that threshold.

When we evaluated 10 individuals from the independent validation cohort, the pattern described above was maintained. The overall performance of WACSAW remained greater than 95% accuracy with low variance and high proportion of the individuals clearing the 95% threshold. This dataset includes the individual for which WACSAW had difficulty identifying sleep according to comparisons with participant recorded log data. Even so, WACSAW outperformed the Actiwatch analysis of the movements over the same period (Table 4). Given that results were derived from the same person at the

**Table 4. WACSAW results compared to results from Actiwatch in percent correct.**

| | Participant id | WACSAW | | | Actiwatch | | |
|---|---|---|---|---|---|---|---|
| | | % correct Accuracy (adj) | % correct Sensitivity (adj) | % correct Specificity (adj) | % correct Accuracy (adj) | % correct Sensitivity (adj) | % correct Specificty (adj) |
| WACSAW | 1 | 95.8 | 94.3 | 96.7 | 87.8 | 79.6 | 92.4 |
| vs | 2 | 98.8 | 98.7 | 98.9 | 92.2 | 92.4 | 92.0 |
| Actiwatch | 3 | 99.5 | 99.5 | 99.5 | 88.9 | 96.0 | 85.4 |
| | 4 | 97.6 | 96.8 | 98.0 | 85.2 | 86.4 | 84.6 |
| | 5 | 95.8 | 97.2 | 95.3 | 73.8 | 89.7 | 67.2 |
| | 7 | 99.4 | 99.2 | 99.5 | 92.2 | 94.0 | 91.4 |
| | 9 | 98.8 | 97.5 | 99.5 | 93.9 | 87.6 | 97.5 |
| | 10 | 97.2 | 94.7 | 98.1 | 80.6 | 84.3 | 79.2 |
| | 16 | 98.8 | 97.2 | 99.7 | 89.5 | 88.9 | 89.9 |
| | 17 | 98.9 | 98.2 | 99.3 | 89.5 | 88.0 | 90.3 |
| | 18 | 99.1 | 99.5 | 98.9 | 87.6 | 91.6 | 85.3 |
| | 19 | 99.6 | 100.0 | 99.4 | 91.9 | 91.4 | 92.0 |
| | 20 | 96.7 | 91.4 | 99.9 | 89.0 | 72.6 | 98.7 |
| | 21 | 97.0 | 98.5 | 96.3 | 90.2 | 80.5 | 94.4 |
| | 22 | 90.0 | 97.2 | 86.5 | 73.4 | 84.2 | 68.2 |
| | Mean | 97.5 | 97.3 | 97.7 | 87.0 | 87.1 | 87.2 |
| | Median | 98.8 | 97.5 | 98.9 | 89.0 | 88.0 | 90.3 |
| | Std Dev | 2.5 | 2.3 | 3.4 | 6.1 | 5.9 | 9.1 |
| | MAD | 1.7 | 1.7 | 2.1 | 4.7 | 4.7 | 7.1 |
| | min | 90.0 | 91.4 | 86.5 | 73.4 | 72.6 | 67.2 |
| | max | 99.6 | 100.0 | 99.9 | 93.9 | 96.0 | 98.7 |
| | % > 95% | 93 | 80 | 93 | 0 | 7 | 13 |

same time period (watches were worn at the same time), a paired test is warranted. While a paired t-test is a candidate, the small sample size may not be normal. Tests for normality would have very low power because of the small sample size and hence we decided to use the Wilcoxon singed rank test [51]. WACSAW performed better across sensitivity, specificity, and accuracy all had a value of $p < 0.0001$ across the 15 samples tested.

We hypothesize that one big difference between the accuracy of WACSAW and the accuracy of the Actiwatch designations appeared to be in classifying quiescent wakefulness. Fig 6 presents 3 instances of large differences between WACSAW and Actigraph. Each of the examples shows the raw MEMS movement recording, how the participant categorized behavior at the time, and how the Actiwatch system classified the movement data. The examples include one participant watching TV (Fig 6A), and two other instances where the Actiwatch classification differed from the participant log during activities such as reading in bed (Fig 6B) and desk work (Fig. 6C), but WACSAW correctly classified them. In each case, WACSAW is able to identify quiet wakefulness behaviors, whereas the Actiwatch algorithm appears to misinterpret it.

Lastly, we compared our outcome to an existing analysis program, GGIR [43,44], that also accepts higher frequency data and has emerged as a useful program for users to evaluate sleep and wakefulness states from actimetry data. The GGIR environment has developed to contain numerous helpful add-ons, but we wanted to compare the core GGIR process to WACSAW. Therefore, we developed a GGIR mimic based on the described process [44] that would deliver data comparable to the output of WACSAW (see Materials and Methods for full details) and benchmarked both to the participant logs. We found that WACSAW was approximately 5% better in overall accuracy, roughly 12% more accurate with sleep, and approximately 2% better with selectivity, which identifies wakefulness (S2 Table). We compared the visual outputs from each program (S1 Fig). GGIR makes substantially more state changes, especially during the night compared to WACSAW. For instance, in participant 19, the first night of sleep is segmented quite frequently using GGIR compared to how WACSAW determines sleep and wakefulness. It may be in these numerous transitions that leads to WACSAW's distinction from GGIR.

It needs to be highlighted at this point that our validation criteria are participant logs. Because it is based on participant recollection and documentation, the logs may reflect longer stable periods of behavior rather than the numerous unrecognized or unrecollected sleep and wakefulness transitions. There is no indication in the logs that there is this number of transitions, but logs are not the definitive determination of state changes. What is apparent from the comparison between GGIR and WACSAW is that they have different tolerances for activity transitions, which may be delivering different types of information to the user. As discussed later, WACSAW needs to be validated using polysomnography (PSG) to determine if the brief movements are indicative of wakefulness. It will also be interesting to determine if those brief wakefulness periods are meaningful to other health or cognitive parameters.

## Discussion

WACSAW is an adaptive sleep-wake classification algorithm with novel applied statistical approaches to segmenting and classifying movement data into sleep and wakefulness. WACSAW delivered an average of >96% accuracy, sensitivity, and selectivity across the entire healthy person cohort when compared to detailed activity logs. Moreover, WACSAW reduced individual error, which resulted in a smaller standard deviation across the entire population. When compared to the clinically-approved Actiwatch, WACSAW had a higher sleep/wake state accuracy and a lower standard deviation between individuals within our participant population, indicating better and more personalized performance in categorizing sleep and wakefulness.

We developed WACSAW from data collected using the GENEActiv actimeter, which uses the MEMS to detect changes in movement at a subsecond sampling rate. The sampling property has distinct advantages in classifying sleep and wakefulness states compared to traditional piezoelectric methods. First, we could compose a more detailed representation of movement variability within a smaller time window, which allows for a faster determination of segmentation boundaries between two periods of different activity levels. Second, we could generate probability distributions within

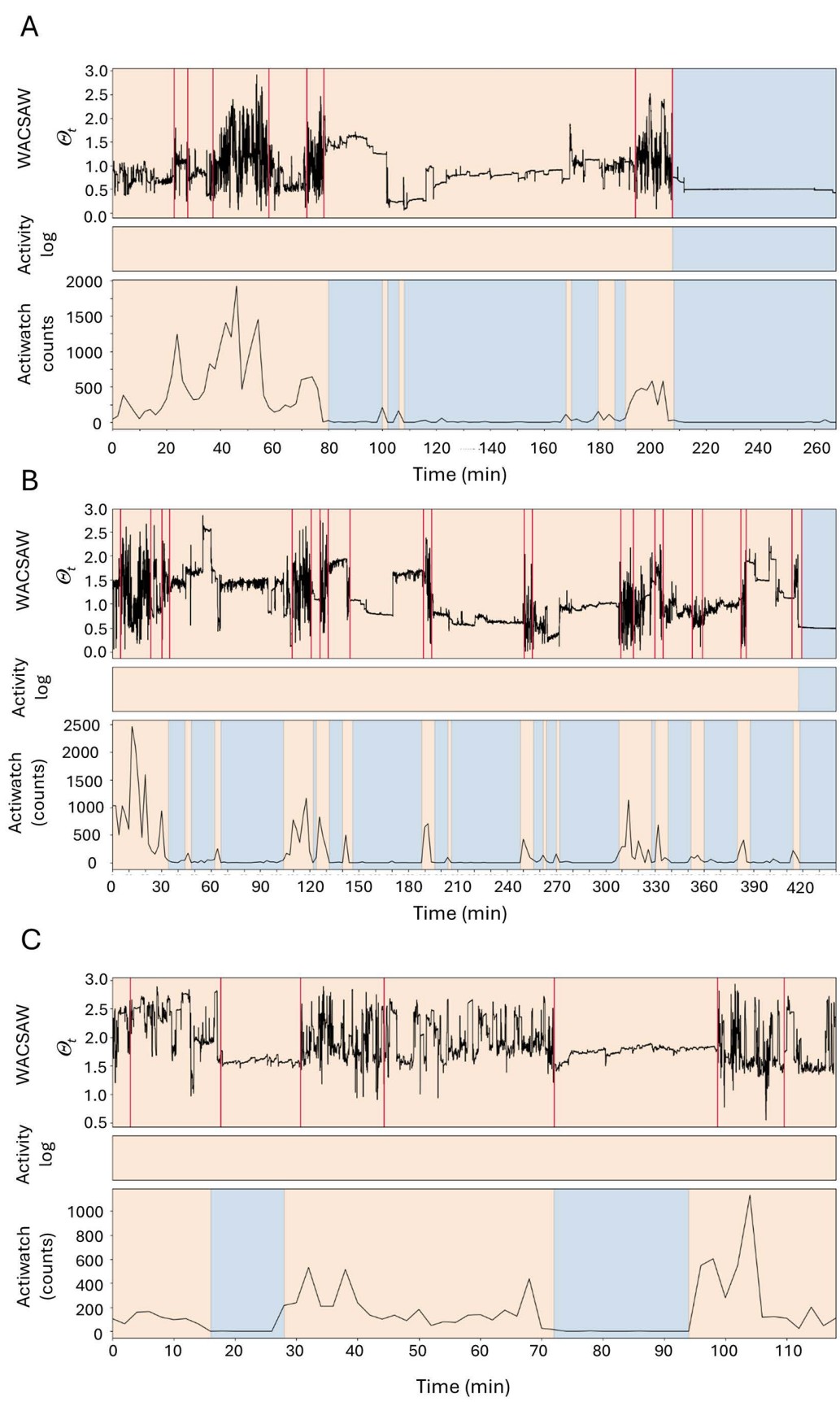

**Fig 6. Comparison of behavioral categorization between WACSAW and Actiwatch.** We chose 3 raw data examples (A-C) to illustrate differences in the performance of WACSAW compared to Actiwatch with a particular focus on the challenges of quiet wakefulness. Orange background indicates a wakefulness categorization by WACSAW or the Actigraph algorithm or that the person logged they were awake (middle bar in each panel. A blue background indicates sleep for all three. In (A) the participant was watching TV in their couch during the wakefulness period leading up to sleep. In (B), a second participant was reading in bed during this period. While in (C), a third participant was working at their desk during the entire recording period shown. Vertical red lines in the WACSAW trace indicate segmentation breaks determined by the algorithm. The Actigraph assigns many more sections as sleep when the participant was engaged in quiet waking activities (see middle bar for actual activity).

smaller time windows. Thirdly, higher frequency collection provides robust distributions of movement statistics in a short time window, making the Wasserstein distance more accurate in determining changes in underlying distributions. Lastly, segmentation becomes more localized, which minimizes sleep-wake overlap in any given region and provides a more accurate partitioning of sleep and wakeful regions. MEMS technology has permitted us to explore statistically based approaches that accommodates individual differences rather than regression-based approaches. Because WACSAW is agnostic to the exact nature of the data, it could also be applied to other classification systems that rely on variability changes.

Another benefit of the WACSAW process is that it produces what we are calling "interim metrics." Interim metrics are outputs that are generated as WACSAW is determining the definition and classification of each of the segments. Examples of these metrics include the Wasserstein time series, the transport energy histograms, and the clustering of the characteristic functions. These interim metrics are unique to the initial data analyzed and could be used to further characterize the sleep and activity over a given period of time or potentially as a measure of confidence for the designations assigned by WACSAW. For example, if the transport energy histograms are overlapping or very closely spaced at a given Wasserstein score, it may indicate poorer sleep/wakefulness segmentation boundaries that introduces errors in interpretation. In such a case, the investigator could dedicate their time and effort to this data set and feel confident that other data sets are categorized well. Another example would be when the shape and overlap of the characteristic functions may indicate poor classification of sleep and wakefulness. In fact, statistical techniques could be derived or applied that would characterize or represent the movement data from the interim metrics. These representations could be a further objective metric to compare with an outcome such as sleep quality or another subjective measure. Additionally, the Wasserstein time series (WTS) could serve as a time series representation of sleep across time. Though summing the first difference of the tilt angle would give us an indication of how much movement occurred during the night, the WTS gives an indication of the relative magnitude of the movement and how long it persisted. Moreover, the Wasserstein series has different shapes for different types of movements. For example, brief movements are more box shaped while longer duration movements are more rounded and have an "m" shape (S2 Fig). These values can then be used in subsequent calculations or time series analyses. We believe that these metrics are a strength of the WACSAW approach and offer an opportunity for more sophisticated and meaningful analyses.

Based on the analyses above, WACSAW offers a different approach and potentially has advantages over current methods, such as regression-based algorithms and neural network approaches. Regression-based approaches worked well for the lower sampling rates of piezoelectric devices which have a strong record for gathering information. Some of the most utilized algorithms are subject to flaws due to the fact that they are regression-based, such as the Sadeh [24] and Cole-Kripke [25]. In fact, a comparison of popular sleep-wake algorithms concluded that for healthy individuals, the Sadeh algorithm still remains optimal [39]. Regression tries to find one equation that best fits the population, which can unknowingly lead to large errors in some individuals that do not conform to the average. The algorithms do have an acknowledged weakness in that sleep sensitivity ranged from 0.86 and 0.99 while specificity performed more poorly and ranged widely between 0.27 and 0.68, likely due to a misclassification of quiet wakefulness, for example desk work, watching television, and reading [26,38].

Additionally, machine learning and deep learning techniques have been applied to actigraphy data, including, including [27–29,31–35]. The most comparable attempts have used a single data modality or single channel data [30,36,52]. In summary, these papers use different sized training populations, different data sources, and applied numerous different approaches to analyze the data. The results demonstrate that there are analyses amongst the methodologies that show improvement over regression techniques. Only some of the analysis methods showed the improvements without a consensus between the approaches and techniques. Another consideration is that it is difficult to determine where the errors in the approaches are. WACSAW demonstrates better numbers with a couple of caveats, WACSAW has not been validated against polysomnography (PSG), which could reduce the overall performance because small sections of the record would show differences. For example, a participant could have woken for a short period of time without movement and not recorded that in their logs, which may bring down WACSAW's overall performance to equivalent levels. On the other hand, WACSAW's approach allows those types of problems to be identified and purposeful changes to the approach to address such issues. In comparison to neural network-based algorithms, WACSAW has many fewer parameters that are also more explainable and that can inform researchers and users about the biology of the system.

WACSAW has aspects within the algorithm that give it properties of having individualized sleep and wakefulness designations. The threshold for designating segment divisions is based on the activity profiles for the two-day span for that individual. Moreover, there are additional opportunities to further adapt WACSAW to individuals or specialized situations, such as sleep disorders, using hyperparameters. For example, increasing the order ($\rho$) in the Wasserstein equation 3 would make it more sensitive to smaller changes in movement distribution. These changes could be used to adapt WACSAW to improve performance in such circumstances, such as the fragmentation that can take place with sleep in aging or sleep disorders. In addition, the program has the potential to be adapted to evaluate different properties depending on the user's interests. For example, expanding the window of the Wasserstein time series around the point of interest would widen the area to detect a change between segments and allow the program to focus on larger time scales, such as those for sleep time and rise time. This change would ignore short-term changes in distribution to preferentially identify larger changes, which would result in fewer segments and support identifying longer overall sleep periods rather than brief awakenings, short naps, or nighttime restroom visits. The steps taken to adapt WACSAW to the individual reduced the variability of accuracy within the population below that seen in the Actiwatch analysis despite the higher accuracy. Thus, WACSAW can be further adapted to specific research or clinical goals.

The approach presented here has limitations, and future investigations may validate or mitigate its generalizability. The most obvious limit is the validation against PSG. Our validation using logs is a practical and opportunistic method that both allowed us to progress on WACSAW but is also a good validation method for many of the characteristics that researchers and participants care about. The weakness is that it does not have the moment-to-moment validation of brain state that can reveal short or brief periods of wakefulness and define sleep latency, which is an important parameter for specific sleep disorders. In fact, we adjusted sleep and wakefulness periods based on a 20 minute window with the acknowledgement that sleep logs can be inaccurate. That said, it only altered accuracy percentages by less than 2%. WACSAW has demonstrated the potential to identify nighttime brief awakenings and potential restlessness in individuals. However, it has thus far been validated to detailed activity logs, thus nighttime states have not been validated against PSG.

Another limitation is the sample size of participants. We were cognizant to include people of different ages, sexes and behavior types. We are aware of the limitations of other algorithms and wanted to specifically challenge WACSAW with these situations to improve the interpretation of activity data. We assessed 22 unique individuals and repeated the experiment on a large subset of those individuals, and WACSAW performed well across all of those individuals. One participant (Part 11) was sick over one night of the test period. It reduced the accuracy seen across the other participants and highlights the unique circumstances that exist in individuals with illness or sleep disorders. We anticipate that alterations described in the paragraph above could optimize WACSAW performance across these situations. Also, the vast majority of our population was affiliated with a university, which may inadvertently skew WACSAW results in an unidentified way.

Lastly at this time, WACSAW does not output some of the typical metrics provided by other algorithms, which is a future goal of this project so that the program is as useful as possible for the sleep community. It is possible that WACSAW could be used in combination with such analysis suites as GGIR to further improve the understanding of how naturalistic sleep impacts health and behavior.

Despite the above drawbacks to this study, we propose that WACSAW may contribute to classifying sleep in naturalistic settings. WACSAW may be more automated in its application than other methodologies. We submit that WACSAW removes the absolute necessity for participant logs in certain types of studies. In fact, we found circumstances where the segmentation points from WACSAW were more reliable than the participant logs because the log times misidentify behaviors because of memory or attention failures from the participants. This may account for a 1–2% discrepancy based on our data. This makes WACSAW more useful in broad population experiments because there is less human intervention required. Additionally, because WACSAW uses only a single stream of data, namely movement data, to classify behavioral states, longer recordings may be possible based on equal battery life and data storage capacities. Moreover, WACSAW classifies varying length segments rather than fixed length epochs, thereby providing more information that could be utilized to improve the interpretation of a sleep or wakefulness call. The interim metrics may offer opportunities to further identify important patterns and relationships within the movement data. They could be used to determine reliability of the classification results based on these interim metrics. WACSAW adapts to individuals by utilizing the Wasserstein threshold and the characteristic function, both derived from the movement data of that specific individual. In contrast to machine learning algorithms, WACSAW is more explainable while maintaining high accuracy. Lastly, WACSAW and its code will be made available to researchers and the public to evaluate and modify to implement in their particular situation.

## Materials and methods

### Experimental design and participants

Participants were recruited by word of mouth and targeted emails. Study coordinators presented written Informed Consent documents, enrolled participants, and provided instructions to conduct the study and research assistants were available to answer any questions. Participants were recruited from July 28, 2020 to July 27, 2021. Upon enrollment, participants completed a demographic survey, including deliberate confirmation that they have no diagnosed sleep disorders and were provided a GENEActiv Original (Activinsights Ltd., Kimbolton, UK) wrist-worn activity monitor [53]. The protocol was approved by the University of Missouri Institutional Review Board (IRB# 2026643).

Participants across all cohorts were aged 18–72 years (average 37.7 ± 14.4 y.o), 55% males, and 77% White (Table 1). We implemented two independent protocols: (1) wear the GENEActiv alone and keep a detailed log of activities and other conditions, such as light levels and sleepiness or (2) simultaneously wear the GENEActiv and Philips Respironics Actiwatch 2 (Cambridge Neurotechnology Ltd., Cambridge, UK and MiniMitter, Respironics Inc., Bend, Oregon, USA) for two days and keep a similar detailed activity log as in the first protocol. There was a subset of individuals that participated in both protocols as well as some that only completed one or the other. Participants were instructed to wear the watch(es) on their nondominant wrist 24 hours a day. The activity log was completed in either a Qualtrics survey accessed through a mobile device or by maintaining an Excel spreadsheet. Participants were asked to reflect on activities at least 2 times per day but encouraged to report as often as possible to increase accuracy in the reports. Surveyed data included start time of activity, end time of activity, activity level, alertness, and ambient light conditions. Temperature and activity measurements from the GENEActiv actimeter were used to validate watch-on compliance.

### Validation of WACSAW classification

Movement data from the GENEActiv watch was collected at 10 Hz frequency and downloaded with the ActiveInsights software. Force measures were gravity compensated according to manufacturer specifications. After WACSAW classification,

performance was compared against participant activity logs using 1 second epochs. Validation metrics included: Accuracy (% of correct classifications over a given time period); Sleep Sensitivity (% correctly classified sleep over the total labeled sleep); and Sleep Specificity (% correctly classified wake over the total labeled wakefulness).

One concern with detailed activity logs is that they can be subjective and reflect an estimate of start and stop points rather than exact times of activity transitions [54]. The human error can impact the validation metrics. Thus, we compared WACSAW output to participant logs using two methods. The first uses the raw activity log data provided by the participant (referred to as 'Raw' log comparison). The second involved a research assistant to manually adjust the start and end times subject to the following criteria: there was a sudden change in volatility of a participant's movement within 20 minutes of the participant indicating there was a change in activity. In such circumstances, the start and endpoint of the activity were adjusted to this timepoint. Adjustments were also made if a participant noted an activity during a reported interval, but no precise time was given. For example, if they stated they got up to check on an animal in the night without providing a time, we assumed a small period of movement during that night represented their logged activity. We refer to these records as "adjusted logs" and these are used to validate WACSAW assignments independently from those based on raw logs. Both sets of validated results are presented, though there are only minor differences between the results.

## Comparison to GGIR

Our study includes a comparison with the method introduced by van Hees et al. called GGIR [44], designed to classify sedentary behavior. This method operates on a time series of tilt angles, which represent movement activity. Posture changes are defined as instances when the tilt angle exceeds a certain threshold, indicating a significant shift in the wearer's position. GGIR uses predefined time and angle thresholds to identify these posture changes. This process creates temporal clusters of activity.

We used the following criteria to create the GGIR mimic. Whenever two significant posture changes occur within a time period shorter than the predefined time threshold, they are considered part of the same active period, and if these shifts are closer in time than the specified threshold, the segments are merged together, akin to linking activity segments and indicating continued activity. A posture shift is defined based on the instantaneous intensity of the movement, contrasting with WACSAW, which analyzes local volatility. When more than one such posture change is detected, the method identifies periods where these changes occur less frequently than the specified time threshold. If such periods exist, they are classified as periods of no significant posture change, implying sedentary behavior. The selection of time and angle thresholds are crucial hyperparameters in this approach. van Heese et al. [44] found that a time threshold of 5 minutes and a tilt angle threshold of 5 degrees generally yielded the most reliable results. For our comparison, we have used these same values [45]. The specific code used is presented in the supplemental material.

## Supporting information

**S1 Appendix. Text describing the hyperparameter grid search.**
(DOCX)

**S1 Table. Top 10 results, based on accuracy, from the hyperparameter sweep.**
(TIF)

**S1 Fig. Comparison of time series of sleep/wakefulness output from WACSAW and GGIR.** We directly compared the output from 3 participants, Participant 5 (A), which was an older individual, Participant 11 (B), in which WACSAW demonstrated its worst performance, and Participant 19 (C), in which WACSAW did very well. An orange part of the trace indicates that the given software called the section wakefulness and a blue portion of the trace indicates that the software labeled the section as sleep. A red background indicates a mismatch between the log and the software results.
(TIF)

**S2 Table. Comparison between WACSAW and GGIR performance.**
(TIF)

**S2 Fig. Example of movement that does not result in segmentation.** (A) A 7-hour segment of the FD of the tilt angle was chosen to highlight the impact of short duration movements at night. In this trace, there are multiple segments identified by WACSAW with boundaries designated by the vertical red lines. Several brief movements, deflections both up and down in the tilt angle (arrows) are not identified as segments. (B) The Wasserstein score time series from the data in (A). The score does not cross the horizontal threshold line to be designated as a segmentation point (arrows correspond to the arrows in (A)). Thus, the continuity of the segment remains in WACSAW analysis. WACSAW does detect two different movement segments between 8 and 9 AM in this example.
(TIF)

## Acknowledgments

The authors would like to thank the participants for volunteering their time and effort to make this study possible. We would like to thank Don Wunsch II at Missouri S&T for resources and valuable foundational discussions. We acknowledge Amber Comfort for her work in preprocessing data for log validation. We also thank Ariel Don for his hard work in creating GUI version of WACSAW and his efforts optimizing code. Lastly, we thank Landon Oelschlaeger for WACSAW validation and troubleshooting the program.

## Author contributions

**Conceptualization:** Austin Vandegriffe, V.A. Samaranayake, Matthew S. Thimgan.

**Data curation:** Austin Vandegriffe, V.A. Samaranayake, Matthew S. Thimgan.

**Formal analysis:** Austin Vandegriffe, V.A. Samaranayake, Matthew S. Thimgan.

**Funding acquisition:** V.A. Samaranayake, Matthew S. Thimgan.

**Investigation:** Austin Vandegriffe, V.A. Samaranayake, Matthew S. Thimgan.

**Methodology:** Austin Vandegriffe, V.A. Samaranayake, Matthew S. Thimgan.

**Project administration:** Matthew S. Thimgan.

**Software:** Austin Vandegriffe, V.A. Samaranayake.

**Supervision:** V.A. Samaranayake, Matthew S. Thimgan.

**Validation:** V.A. Samaranayake.

**Writing – original draft:** Austin Vandegriffe, V.A. Samaranayake, Matthew S. Thimgan.

**Writing – review & editing:** Austin Vandegriffe, V.A. Samaranayake, Matthew S. Thimgan.

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
