## [Decision Letter · Decision Letter 0]

30 Jun 2025

Dear Dr. Thimgan,

Thank you for submitting your manuscript to PLOS ONE. After careful consideration, we feel that it has merit but does not fully meet PLOS ONE’s publication criteria as it currently stands. Therefore, we invite you to submit a revised version of the manuscript that addresses the points raised during the review process.

We look forward to receiving your revised manuscript.

Kind regards,

Julio Alejandro Henriques Castro da Costa

Academic Editor

PLOS ONE

Journal Requirements:

3. In the online submission form, you indicated that your data will be submitted to a repository upon acceptance. We strongly recommend all authors deposit their data before acceptance, as the process can be lengthy and hold up publication timelines. Please note that, though access restrictions are acceptable now, your entire minimal dataset will need to be made freely accessible if your manuscript is accepted for publication. This policy applies to all data except where public deposition would breach compliance with the protocol approved by your research ethics board. If you are unable to adhere to our open data policy, please kindly revise your statement to explain your reasoning and we will seek the editor's input on an exemption.

5. Please remove all personal information, ensure that the data shared are in accordance with participant consent, and re-upload a fully anonymized data set.

Additional guidance on preparing raw data for publication can be found in our Data Policy (https://journals.plos.org/plosone/s/data-availability#loc-human-research-participant-data-and-other-sensitive-data) and in the following article: http://www.bmj.com/content/340/bmj.c181.long .

Reviewers' comments:

Reviewer's Responses to Questions

**Comments to the Author**

1. Is the manuscript technically sound, and do the data support the conclusions?

Reviewer #1: Partly

2. Has the statistical analysis been performed appropriately and rigorously?

Reviewer #1: Yes

3. Have the authors made all data underlying the findings in their manuscript fully available?

Reviewer #1: Yes

4. Is the manuscript presented in an intelligible fashion and written in standard English?

Reviewer #1: Yes

Reviewer #1: The manuscript describes WACSAW, a method that uses segmentation and classification to label high-frequency actimetry data as either sleep or wake. WACSAW relies on Wasserstein distances and k-nearest-neighbor clustering of certain features. The method was tested on two small groups: six people for development and sixteen people as an independent set. According to the manuscript, WACSAW achieved over 95% median accuracy per epoch and was about 10% more accurate than a commercial device (Philips Respironics Actiwatch), especially for quiet wakefulness. This is an important topic since accurate, understandable, and device-independent sleep scoring is lacking in real-world settings. The manuscript is mostly clear and provides good details about the methods.

However, there are some significant points to address:

There is no gold-standard validation. All results are compared with participant activity logs, not with polysomnography (PSG), which is the standard for sleep studies. The authors mention this limitation, but without PSG data, we cannot know how well WACSAW detects short arousals, sleep latency, or distinguishes REM from NREM sleep. At least some PSG data from a subset of subjects should be provided.

External validation would help confirm reliability.

The manuscript does not assess real-time feasibility.

The method for selecting thresholds is not well explained; it needs to be clear that these were not chosen “by eye.”

More thorough statistical comparison with Actiwatch, including proper hypothesis testing, is needed.

**Do you want your identity to be public for this peer review?** For information about this choice, including consent withdrawal, please see our Privacy Policy

Reviewer #1: **Yes: ** Prof. Dr. Murat Ozgoren, MD PhD

---

## [Author Response · Author response to Decision Letter 1]

13 Aug 2025

We appreciate the editor’s and reviewer’s time and effort in evaluating the manuscript. We have addressed each of the points raised by both. The responses are presented below.

Journal Requirements:

We have updated the manuscript to the formatting described in the above documents.

We have updated our licensing agreement to CC BY-NC-SA for use and adaptation throughout the sleep community.

3. In the online submission form, you indicated that your data will be submitted to a repository upon acceptance. We strongly recommend all authors deposit their data before acceptance, as the process can be lengthy and hold up publication timelines. Please note that, though access restrictions are acceptable now, your entire minimal dataset will need to be made freely accessible if your manuscript is accepted for publication. This policy applies to all data except where public deposition would breach compliance with the protocol approved by your research ethics board. If you are unable to adhere to our open data policy, please kindly revise your statement to explain your reasoning and we will seek the editor's input on an exemption.

The data and code have now been uploaded to the doi: https://doi.org/10.71674/mq6j-z250

Austin Vandegriffe’s ORCID ID is 0009-0008-8623-6589 and has been entered into the system. I was unable to enter his ORCID ID along with mine.

5. Please remove all personal information, ensure that the data shared are in accordance with participant consent, and re-upload a fully anonymized data set.

The data and code have now been uploaded to the doi: https://doi.org/10.71674/mq6j-z250

Reviewers' comments:

Reviewer's Responses to Questions

Comments to the Author

1. Is the manuscript technically sound, and do the data support the conclusions?

Reviewer #1: Partly

2. Has the statistical analysis been performed appropriately and rigorously?

Reviewer #1: Yes

3. Have the authors made all data underlying the findings in their manuscript fully available?

Reviewer #1: Yes

4. Is the manuscript presented in an intelligible fashion and written in standard English?

Reviewer #1: Yes

5. Review Comments to the Author

Reviewer #1: The manuscript describes WACSAW, a method that uses segmentation and classification to label high-frequency actimetry data as either sleep or wake. WACSAW relies on Wasserstein distances and k-nearest-neighbor clustering of certain features. The method was tested on two small groups: six people for development and sixteen people as an independent set. According to the manuscript, WACSAW achieved over 95% median accuracy per epoch and was about 10% more accurate than a commercial device (Philips Respironics Actiwatch), especially for quiet wakefulness. This is an important topic since accurate, understandable, and device-independent sleep scoring is lacking in real-world settings. The manuscript is mostly clear and provides good details about the methods.

However, there are some significant points to address:

• There is no gold-standard validation. All results are compared with participant activity logs, not with polysomnography (PSG), which is the standard for sleep studies. The authors mention this limitation, but without PSG data, we cannot know how well WACSAW detects short arousals, sleep latency, or distinguishes REM from NREM sleep. At least some PSG data from a subset of subjects should be provided.

We thank the reviewer for their comment. For this study we did not collect PSG data due to space and financial constraints. Therefore, we do not provide these data because they were not collected in this cohort. Based on the results developed and presented in this manuscript, we have initiated collaborations with labs that have access to reliable PSG data collection to answer the questions that the reviewer has posed. Our plan is to continue to address these parameters to develop the most useful tool that can be used by the sleep community.

However, in spite of this shortcoming, we think there is scientific merit in what we have presented. First, we have proposed an algorithm that outperforms two existing methods, albeit this performance edge is based on participant logs. We point out that the Actiwatch 2 has been validated against PSG and is clinically accepted accuracy compared to sleep [1]. Given that our data suggest that WACSAW outperforms this validated technology reinforces that WACSAW is likely a beneficial technology. This demonstrates that the method we use to segment the actigraphy time series as well as the classification strategy that is used have the potential to perform well, possibly with some revisions, when tested on PSG data. Thus, we offer a potential methodology that others with access to PSG data can utilize to build an even more accurate model than the current version of WACSAW. We think that this merits its dissemination.

• External validation would help confirm reliability.

To this point, WACSAW is the only algorithm that has interim metrics that can be used for validation within an individual. Other algorithms do not have this ability. The field can use this to identify which individuals are reliable when compared to external validation. While we agree that external validation can bring additional confirmation of reliability, as pointed out earlier, we stress that our contribution is not merely presenting a more accurate sleep-wake classification algorithm, but also to propose a novel approach to this task that does not depend solely on population metrics that regression based methods do and also provide intermediate metrics that “black-box” type approaches may not provide.

• The manuscript does not assess real-time feasibility.

Real-time implementation is something that is not specifically addressed, though we have given thought to this aspect. At this moment, there is a 15-min future window that is incorporated into WACSAW. This means that there would have to be at least a 15-min delay on any “real-time” analysis, which is not a substantial delay. Moreover, we have been converting code from Python to C++ for faster implementation speeds. Thus, while this is a goal for the future, we don’t believe it is the right time to discuss this aspect quite yet. Instead, we have focused on how WACSAW might be used in “after-the-fact” analysis.

• The method for selecting thresholds is not well explained; it needs to be clear that these were not chosen “by eye.”

We agree with the reviewer that a “by eye” determination of thresholds and hyperparameters is not a reliable methodology to determine these cut-offs. Moreover, what works in one individual may not work in another. Thus, a methodology that adapts to the individual must be developed to remove human intervention as much as possible. To this end, we had included our hyperparameter sweep in the supplementary data to demonstrate which variables and approaches provide the best results in the development cohort. We then go on to show that independent cohorts, that the algorithm was not trained on, show similar results using these empirically derived hyperparameters. These analyses are presented in the hyperparameter section of the methods and the supplemental data. In addition, we do not present our algorithm as the ultimate solution. Our intention is to present our methodology of segmenting and classifying these segments as an alternative approach with potential that other researchers can optimize based on extensive applications to other data sets.

• More thorough statistical comparison with Actiwatch, including proper hypothesis testing, is needed.

We thank the reviewer for this comment. We have now included statistical analyses comparing the results from WACSAW to those of the Actiwatch. The statistical comparison is now included in the text at lines 566-571

“Given that results were derived from the same person at the same time period (watches were worn at the same time), a paired test is warranted. While a paired t-test is a candidate, the small sample size may not be normal. Tests for normality would have very low power because of the small sample size and hence we decided to use the Wilcoxon singed rank test Demsar (2006). WACSAW performed better across sensitivity, specificity, and accuracy all had a value of p<0.0001 across the 15 samples tested.”

that replaces the existing text from that section.

One last note, we realized that one of the tables (Table 2) had a repeated data point for participant 5. The table has been updated to incorporate the appropriate data and the resulting statistics.

---

## [Decision Letter · Decision Letter 1]

14 Sep 2025

WACSAW: An adaptive, statistical method to classify movement into sleep and wakefulness states

PONE-D-25-25743R1

Dear Dr. Thimgan,

We’re pleased to inform you that your manuscript has been judged scientifically suitable for publication and will be formally accepted for publication once it meets all outstanding technical requirements.

Kind regards,

Julio Alejandro Henriques Castro da Costa

Academic Editor

PLOS ONE

Additional Editor Comments (optional):

Reviewer #1:

Reviewers' comments:

Reviewer's Responses to Questions

**Comments to the Author**

Reviewer #1: All comments have been addressed

2. Is the manuscript technically sound, and do the data support the conclusions?

Reviewer #1: Yes

3. Has the statistical analysis been performed appropriately and rigorously?

Reviewer #1: I Don't Know

4. Have the authors made all data underlying the findings in their manuscript fully available?

Reviewer #1: Yes

5. Is the manuscript presented in an intelligible fashion and written in standard English?

Reviewer #1: Yes

Reviewer #1: The authors clearly indicated the "limitations" that I formerly raised:

"There is no gold-standard validation. All results are compared with participant activity logs, not with polysomnography (PSG), which is the standard for sleep studies. The authors mention this limitation, but without PSG data, we cannot know how well WACSAW detects short arousals, sleep latency, or distinguishes REM from NREM sleep. At least some PSG data from a subset of subjects should be provided.

External validation would help confirm reliability.

The manuscript does not assess real-time feasibility."

These points have been presented in the new manuscript

**Do you want your identity to be public for this peer review?** For information about this choice, including consent withdrawal, please see our Privacy Policy

Reviewer #1: **Yes: ** Prof. Dr. Murat Ozgoren, MD PhD

---

## [Editor Report · Acceptance letter]

PONE-D-25-25743R1

PLOS ONE

Dear Dr. Thimgan,

I'm pleased to inform you that your manuscript has been deemed suitable for publication in PLOS ONE. Congratulations! Your manuscript is now being handed over to our production team.

Kind regards,

on behalf of

Dr. Julio Alejandro Henriques Castro da Costa

Academic Editor

PLOS ONE